# The Relationship between Nkx2.1 and DNA Oxidative Damage Repair in Nickel Smelting Workers: Jinchang Cohort Study

**DOI:** 10.3390/ijerph16010120

**Published:** 2019-01-04

**Authors:** Zhiyuan Cheng, Ning Cheng, Dian Shi, Xiaoyu Ren, Ting Gan, Yana Bai, Kehu Yang

**Affiliations:** 1Evidence-Based Medicine Centre, School of Basic Medical Sciences, Lanzhou University, Lanzhou 730000, China; zhiyuan.cheng@yale.edu; 2School of Public Health, Department of Epidemiology and Statistics, Lanzhou University, Lanzhou 730000, China; shid2012@163.com (D.S.); renxy2016@163.com (X.R.); gant17@lzu.edu.cn (T.G.); Baiyana@lzu.edu.cn (Y.B.); 3Centre of Medical Laboratory, School of Basic Medical Science, Lanzhou University, Lanzhou 730000, China; Chengn@lzu.edu.cn

**Keywords:** nickel smelters, Nkx2.1, pSmad2, oxidative stress, DNA repair

## Abstract

*Background*: Occupational nickel exposure can cause DNA oxidative damage and influence DNA repair. However, the underlying mechanism of nickel-induced high-risk of lung cancer has not been fully understood. Our study aims to evaluate whether the nickel-induced oxidative damage and DNA repair were correlated with the alterations in Smad2 phosphorylation status and Nkx2.1 expression levels, which has been considered as the lung cancer initiation gene. *Methods*: 140 nickel smelters and 140 age-matched administrative officers were randomly stratified by service length from Jinchang Cohort. Canonical regression, *χ*^2^ test, Spearman correlation etc. were used to evaluate the association among service length, MDA, 8-OHdG, hOGG1, PARP, pSmad2, and Nkx2.1. *Results*: The concentrations of MDA, PARP, pSmad2, and Nkx2.1 significantly increased. Nkx2.1 (*r*_s_ = 0.312, *p* < 0.001) and Smad2 phosphorylation levels (*r*_s_ = 0.232, *p* = 0.006) were positively correlated with the employment length in nickel smelters, which was not observed in the administrative officer group. Also, elevation of Nkx2.1 expression was positively correlated with service length, 8-OHdG, PARP, hOGG1 and pSmad2 levels in nickel smelters. *Conclusions*: Occupational nickel exposure could increase the expression of Nkx2.1 and pSmad2, which correlated with the nickel-induced oxidative damage and DNA repair change.

## 1. Introduction

Nickel is an essential metal element that widely exists, however high exposure of nickel causes genotoxicity, immunotoxicity, mutagenicity, and carcinogenicity [1]. In 1976, the International Agency for Research on Cancer (IARC) classified nickel compounds such as nickel carbonyl and nickel chloride as group I human carcinogens [2]. Recent epidemiology studies provided supporting evidence that occupational nickel workers have much higher incidence rates of lung cancer and nasopharynx cancer in comparison to those without the exposure [3]. Our early studies also showed that the age-standardized lung cancer mortality rate in Jinchang Cohort was two-times higher than the national mortality rate in China and three-times higher than the local general population mortality rate [4,5]. The underlying mechanisms of nickel-induced carcinogenicity, however have not been fully understood.

A new hypothesis was proposed in 2006 that the reactivation of embryonic lung development genes could lead a core factor during the initiation stage of lung cancer [6,7], the over-expression of Nkx2.1 gene could reactivate an early fetal gene expression pattern which could leading to tumor growth [8]. Previous studies further verified this hypothesis that Nkx2.1 gene prominent amplification in primary lung cancer tissue samples, highly suggested that the reactivation of Nkx2.1 has a close association with the initiation stage of lung cancer [9,10,11], Regulated by the TGF-β signaling pathway, Nkx2.1 is a homeodomain-containing transcription factor, solely encoded by Nkx2.1 gene belonging to the Nkx superfamily. Nkx2.1 is responsible for inducing fetal lung morphogenesis, preserving embryonic lung cell differentiation, and maintaining the normal function of embryo lung tissue during the fatal lung development stage [12]. Nevertheless, recent studies have also suggested that Nkx2.1 is only expressed in lung adenocarcinoma, lung small cell cancer, and thyroid cancer, which highly supports that Nkx2.1 could be a potential biomarker for lung cancer diagnosis [13,14]. Moreover, studies further proved that the amplification and over-expression of Nkx 2.1 highly contributing to lung cancer cell proliferation rates and survival which implicate Nkx 2.1 as a lineage-specific oncogene in lung cancer [15,16,17]. Encoded by Smad genes, members of the Smad protein family played an important signal transportation role in TGF-β signal pathway [18,19,20,21]. Since phosphorylation of Smad2 is the key step of the Smad signaling pathway, we chose phosphorylated Smad2 as a biomarker to measure the activation of the TGF-β signaling pathway. Phosphorylated by high-level Activin-A and separated from SARA (Smad anchor for receptor activation) protein, pSmad2 can be combined with Smad4 to form a complex that directly binds with the Nkx2.1 gene promoter in the nucleus and activates the expression of Nkx2.1 [22,23]. As an up-stream regulator signaling pathway of Nkx2.1 gene, the excessive expression of TGF-β could indirectly reflects the activation level of Nkx2.1 and contribute to tumorigenesis.

A potential mechanism of nickel exposure-induced carcinogenicity is through oxidative stress [24]. In our study, malondialdehyde (MDA) and 8-Oxo-2′-deoxyguanosine (8-OHdG) were chosen as the biomarkers to evaluate the level of Lipid Peroxidation (LPO) and DNA oxidative stress. Furthermore, human 8-oxoguanine DNA N-glycosylase 1 (hOGG1) and Poly ADP-ribose polymerase (PARP) that specifically recognize and repair 8-oxog were chosen to evaluate the level of DNA repair ability. Based on our former study, occupational nickel exposure could significantly affect the expression of 8-OHdg and hOGG1 [25]. However, the underlining mechanism were still under poorly understanding. Our study aims to further investigate the influence of occupational nickel exposure on oxidative stress, DNA repair, and Nkx2.1 expression. Correspondingly to investigate whether nickel exposure induced oxidative stress and DNA repair capability could influence the expression of Nkx2.1 gene or not, which subsequently increase lung cancer risk in nickel smelters. 

## 2. Materials and Methods

### 2.1. Subjects

Approved by the Ethics Committee of Lanzhou University and performed according to the ethical standards of The Declaration of Helsinki formulated in 1964, all participants provided written informed consent before enrolling into the study. The study population was selected from the world’s largest multi-metal occupational exposure cohort: Jinchang Cohort. 48,000 participants were enrolled in the baseline data between June 2011 to August 2015 from the Jinchuan Nonferrous Metals Corporation (JNMC), which is the largest nickel producer in China [26]. Eligibility criteria include age of 18 years old or older, to have completed a physical examination, and donated blood and urine samples at baseline. Performed by uniformly trained interviewers, all participants were interviewed using a standardized questionnaire that includes information on occupational history, lifestyle factors, medical history, and family history. 

### 2.2. Nickel Smelter Group Sampling

From our former multiple urinary metal studies [27], we have found that nickel smelters who worked in the nickel-smelting factory have the highest internal nickel concentration (Table 1). Thus, we randomly cluster sampled 140 male smelters (aged 20–55, mean age ± standard deviation: 38.31 ± 8.91) from 969 smelters as nickel exposed group who were only ever been employed as nickel smelter group (age and service length distribution is shown in Appendix A). To eliminate confounding effects of age on the relationship between length of service and nickel exposure, a diagonal sampling method was used to select 20 subjects for each 5-year service length subgroups (service length of 0–4, 5–9, 10–14, 15–14, 15–19, 20–24, 25–29, and above 30 years). (sampling results are shown in Appendix A). Participants’ age and employment length were highly correlated (*r*_s_ = 0.980, *p* < 0.0001). 

### 2.3. Administrative Officer Group Sampling

Each participant in nickel smelters group was matched to a cohort member who had only been employed by the administrative office of Worker’s Hospital of Jinchang Non-ferrous Company and Jinchang Railway Transport Company, who had the lowest nickel exposure [27]. Since smoking, alcohol consumption, and tea drinking are associated with oxidative stress [28], these variables were also included as matching variables in addition to age (within 3 years) and employment length (5-year interval). All controls had no diagnosis of chronic diseases such as cancer, cardiovascular diseases, and diabetes and had no recent medication use. Considering the impact of genetic factors and age differential in each service length categories, all the participants in nickel smelters group and administrative officers group were selected with same component ratio of cancer family history. The distributions of matching variables between the exposed and non-exposed groups were similar (Table 2 and Table 3).

### 2.4. Detection Method

Fasting blood samples were collected under aseptic conditions in the morning and split into four 3 mL aliquots (two anticoagulants and two non-anticoagulants) and stored at −80 °C. 6 different biomarkers were measured with standard procedures. Our anterior results suggested that occupational nickel exposure could significantly increase the expression of 8-OHdg, the expression of hOGG1, however, was significantly inhibited by nickel exposure. The detail measurement methods of malondialdehyde (MDA), 8-hydroxy-2’-deoxyguanosine (8-OHdG), and human 8-hydroxyguanine DNA-glycosylase 1 (hOGG1) were described previously [25]. 

Serum poly(ADP-ribose) polymerase (PARP) concentrations were measured using a double antibody sandwich method (Prod. No. 1B935, RayBiotech, Inc., Norcross, GA, USA,). 10 μL of serum was added to the microtiter plate with 40 μL diluent and incubated at 37 °C for 30 min. After washing for five times with scrubbing solution, then added 50 μL of HRP-conjugated detection antibody to each microtiter, and incubated for 30 min at 37 °C. Following five times washing, incubated again in a dark environment for 15 min at 37 °C after adding 50 μL of chromogenic agent A and B. Finally, the reaction was stopped by the addition of 50 μL stop buffer and recorded at 450 nm using an Automatic Enzyme Mark Analyzer (Powerwave X, Bio-TeK Instruments, Inc., Winooski, VT, USA). 

Nkx2.1 and phospho-Smad2 (pSmad2) were detected with a biotinylated double antibody sandwich enzyme-linked immunosorbent assay (prod. No. 20120720, Jingtian Biotechnologies Co. Ltd., Shanghai, China). Serum samples were loaded at 20 μL into a microtiter plate which coated with a purified Nkx2.1/pSmad2 antibody. After incubation, the antibodies (5 μL) that remained bound to the Nkx2.1/pSmad2 in the microtiter plate were further bound with the HRP–conjugated secondary antibody (25 μL). Then, the plate was incubated at 37 °C for 60 min. After extensive washing for 5 times and desiccated, 25 μL of *TMB* and *HRP* were added into each well, under the protection from light at 37 °C for 10 min, 25 μL of stop buffer was added in the end. At a wavelength of 450 nm, the absorbance was measured by the Automatic Enzyme Mark Analyzer (Powerwave X, Bio-TeK Bio-TeK Instruments Inc., Winooski, VT, USA). 

### 2.5. Statistical Analysis

SAS 9.4 (SAS Institute, Cary, NC, USA) was used to analyze the data, and EXCEL 2013 (Microsoft Corporation, Redmond, WA, USA) and SigmaPlot 12.5 (Systat Software Inc., San Jose, CA, USA) were used to create figures. Behavioral factors were compared between exposed and non-exposed groups using the *χ*^2^ test. Protein expression and oxidative stress biomarkers were compared between exposed and non-exposed groups using the Mann-Whitney U test. ANOVA tests were used to examine the differences in biomarkers among different service lengths. Spearman rank correlation was used to calculate correlations between different biomarkers and service lengths. 

Canonical correlation, a multivariate statistical analysis used to explore the correlation between two groups of varieties by means of principal component analysis. Considering the multifaction of lung cancer, the canonical correlation model was employed among nickel smelters to explore whether the re-activation of embryonic lung development gene was correlated with the nickel-induced DNA damage and repair changing or not. Seven different biomarkers including exposure time variables was composed of two groups according to their own attributes. The group *V* composed of 5 different independent varieties including exposure time, MDA, 8-OHdG, hOGG1, and PARP to measure the level of DNA damage and repair. The group *W* was composed of two independent varieties that include Nkx2.1 and pSmad2, to evaluate the expression level of embryonic lung development genes. All analyses were a two-sided test and *p* = 0.05 was set as the statistical significance threshold.

## 3. Results

### 3.1. Oxidative Damage (MDA) Differences between the Nickel Smelter Group and Administrative Officer Group

The concentration of MDA was positively correlated with increased service length in both groups (Table 4). However, the concentration of MDA in the nickel smelter group was significantly higher than that in the administrative officer group (*p* < 0.001). The concentration of PARP in the nickel smelter group was also significantly higher than that in the administrative officer group (*p* < 0.001). The expression of PARP reached a peak after exposure to nickel for 10–14 years and then demonstrated a declining trend with increased exposure time (*r*_s_ = −0.175, *p* = 0.039) in the nickel smelter group, while the expression of PARP was negatively correlated with employment length increase (*r*_s_ = −0.376, *p*< 0.001) in the administrative officer group (Table 4).

### 3.2. Expression of Nkx2.1 and pSmad2 Proteins between the Nickel Smelters and Administrative Officers Groups

Within every employment length interval, the concentrations of Nkx2.1 and pSmad2 in the nickel smelter group were significantly higher than those in the administrative officer group (*p* < 0.001). Nkx2.1 (*r*_s_ = 0.312, *p* < 0.001) and pSmad2 (*r*_s_ = 0.232, *p* = 0.006) were positively correlated with the employment length in the nickel smelter group. The concentration of pSmad2 among the nickel smelter group indicated a sharply increased trend after 5–9 years of service length and then started to decrease (Figure 1), while negative correlations were observed between pSmad2 and service length for the administrative officer group (*r*_s_ = −0.169, *p* = 0.046, Table 5). The Nkx2.1 concentration between the two groups was inconspicuous under 5 years of employment. While after 5 years of occupational nickel exposure, the concentration of Nkx2.1 in nickel smelter group was positively correlated with service length increasing till 20–24 years of employment (*r*_s_ = 0.312, *p* ≤ 0.001, Table 5, Figure 2). No significant correlation between service length and the concentration of Nkx2.1 was observed in the administrative officer group (*r*_s_ = −0.146, *p* = 0.085, Table 5, Figure 2).

### 3.3. Canonical Correlation Analysis of DNA Damage, Repair, and Embryonic Lung Development Biomarkers

The results specified that the correlation between the two groups (group V and group W) was statistically significant (*p*_1_ < 0.0001). The independent group variables with a significant contribution to the correlation coefficient of embryonic development gene expression were: 8-OHdG, hOGG1, PARP, and employment length, but not MDA. The expression of embryonic lung development gene is positively correlated with the nickel exposure time, 8-OHdG, and PARP, negatively correlated with the expression level of hOGG1. The canonical correlation equation to represent the group inner correlation built by standardized canonical coefficient 1 (Table 6) is as follows:$$\{\begin{array}{l} V_{1}=0.278X_{exposure\;time}+0.037X_{MDA}+0.764X_{8-OHdG}+0.586X_{PARP}-0.453X_{hOGG1}\\ W_{1}=0.989Y_{Nxk2.1}-0.008Y_{pSmad2}\;\left(R2=0.518\right) \end{array}$$

## 4. Discussion

### 4.1. Occupational Nickel Exposure Could Enhances the LPO Processes and DNA Oxidative Damage

After excluding the confounding factors such as smoking, alcohol drinking, etc., over-produced MDA is involved in the molecular mechanism of nickel toxicity in smelters when compared with administrative staff [29]. MDA is the most represented terminal metabolite of LPO, which not only reflects the content of free radicals generated in the body but also can predict the intensity and rate of LPO reaction and cellular oxidative damage [30]. Our study confirms that the concentration of MDA increased along with aging both in nickel smelters group and administrative officer group. However, the concentration of MDA in the nickel smelter group is statistically higher than the administrative officer group in every service length category. This result suggests that with the increase in nickel exposure time, nickel could accumulate in the human body and continuously enhance the oxidative process of polyunsaturated fatty acids in the cell membrane and exacerbate the extent of LPO [31]. 

For DNA oxidative stress, since the similarity of the spatial structure between 7, 8-dihydro-8-oxoguanine (8-Oxog) and adenine, consequently the oxidized guanine base-pair can be easily misread from G:C to T:A [32]. 8-Oxog can be hydrolyzed into 8-OHdG, which is widely distributed in urine, blood and tissue fluids. Based on the abundance and stability of 8-OHdG reflected in both exposure and effect level, it is a putative indicator for intracellular DNA oxidative damage [33,34]. Our former research [25] reported that no correlation between service length and DNA oxidative damage was observed in the administrative officer group. However, the concentration of 8-OHdG in the nickel smelter group was significantly higher than that in the administrative officer group for each employment length category. Moreover, the concentration of 8-OHdG was significantly elevated among nickel smelters, particularly after exposure to nickel for 10–14 years. This result further suggested occupational nickel exposure could simultaneously cause DNA oxidative damage which peaked at 10–14 years of service length [35,36] (Appendix A). 

### 4.2. Nickel Exposure Could Decrease the Expression of hOGG1 but Amplifies the Expression of PARP

8-Oxoguanine DNA glycosylase (OGG1) is a bifunctional enzyme harboring N-glycosylase and β-lyase and encoded by the 8-hydroxyguanine DNA glycosylase 1 gene (hOGG1) [37,38]. It mainly identifies and repairs the oxidative stress via DNA glycosylase modifications by its protein-protein interactions to regulate the base excision repair pathway (BER) [39]. Specifically, for 8-oxoG, 8oxoA, and FapyG lesions specifically recognized by OGG1, our former research [25] showed that the expression level of hOGG1 in the administrative officer group was significantly higher than that in the nickel smelter group for every employment length category (*p* < 0.001) which suggests that nickel exposure decreased hOGG1 expression and subsequently increased oxidative stress. No correlation was observed between the expression levels of hOGG1 and service length in both nickel smelter and administrative officer groups which indicated the expression of *hOGG1* decreased by nickel exposure but was not related to aging [40]. This may further refer that aging is not the decisive factor for hOGG1 expression and activity in this nickel exposure population. The reduced hOGG1 expression should be the reason of 8-OHdG accumulated spontaneously in the nickel smelters group [41]. The study indicated that the expression of hOGG1 had strong organ specificity [42]. Also, hOGG1 deficient mice will spontaneously develop into lung carcinoma or adenoma after being born for 1.5 years [43]. These results may further suggest that the decreased glycosylase mediated 8-oxog excision may have biologically relevant with the initiate of lung cancer. (Appendix A)

PARP interacts with OGG1 to form a complex, which is a key factor in the BER pathway [44]. The insufficiency of OGG1 could impair the activity of PARP and disrupt the catalytic activation of this complex. In consideration of OGG1 being the precursor of PARP related post-translational modification in the BER process and the nucleotide excision repair pathway, the oxidative stress could recede the interaction of the binding multiprotein complexes [45]. The OGG1 downward regulation should also decrease the expression of PARP. However, our study demonstrated a contrary result that the expression of PARP in nickel exposure group was significantly higher than the officers’ group. The concentration of PARP in nickel smelter group peaked at the 10–14 services length category, and both nickel smelter group and office group showed a decreased trend along with the growth of service length. We speculate that nickel compounds may activate the expression of PARP during other signal pathway despite the receding of hOGG1. Importantly, the decrease of OGG1 catalytic activity may hinder the activation of PARP and will compensatively increase the expression of PARP in vivo to respond to the loss of PARP activity [46,47]. Meanwhile, phagocytosed Ni^2+^ can competitively replace Mg^2+^, attenuating PARP repair activity and further enhance oxidative damage. We hypothesize that the subsequently increased PARP expression may be a response to the loss of PARP activity [48]. However, the mechanism of nickel compounds blocked the expression of hOGG1 still remains in further study. 

Additionally, recent studies found that PARP inhibitor is a promising and controversial synthetic lethal therapy for cancer patients with specific mutation [49]. Clinical trials have shown that the platinum-based (Carboplatin) therapy could safely and effectively enhance the therapeutic effect of PARP inhibitor [50]. Similarly, studies have shown that nickel exposure could significantly decrease the expression of hOGG1 and increase the PARP sensitivity to PARP inhibitors [51], suggesting that nickel-based derivatives could be a potential PARP inhibitor adjuvant therapy to enhance its efficacy during the cancer treatment. 

### 4.3. Nickel Exposure Could Significantly Increase the Expression of Nkx2.1 and pSmad2; Oxidative Stress and DNA Repair Capability Were Correlated with the Expression of Nkx2.1 and pSmad2

Smad2 protein is typically present in the cytoplasm in monomer form, and plays an indispensable role in the development of embryonic tissue. Only phosphorylated Smad2 protein can transport into the nucleus and by binding with gene promoter, it participates in the regulation of lung interstitial cell differentiation and influences the maturation and function establishment of alveolar cells [52]. The downstream of the TGF-β signaling pathway mainly consist of three regulable regions, which include Activator protein-1 (AP-1), Nuclear factor-κB (NF-κB) and TTF-1/Thyroid transcription factor 1 (Nkx2.1). Among all of them, AP-1 is a transcriptional activator, but also a receptor for the inner stimulating receptor and especially for respiratory epithelial cells proliferation [53]. Simultaneously, NF-κB is an important transcription factor for host immune defense. The increased NF-κB expression can enhance the expression of Vascular Epithelial Growth factor (VEGF), proteolytic enzymes and other genes related to the invasion and metastasis of malignant cells [54]. Studies found that nickel compounds (eg. nickel chloride) significantly activated TGF-β signaling pathway by TLR4/MyD88 and NF-κB signaling pathway, and further enhance the invasive ability of lung cancer tumor cells [55,56]. Since the TGF-β superfamily has a comprehensive biological activity and especially during the early embryonic development and cell apoptosis [57], any abnormality in this pathway can lead to signal transduction disorder, inhibit the growth of respiratory epithelial cells, and promote the occurrence and development of cancer. Our results indicated that occupational nickel exposure increased pSmad2 expression level. The pSmad2 concentration analysis shows no significant difference in the first two service length intervals. While starting from the third service length interval, the concentration of pSmad2 in the nickel smelter group begin rising dramatically. Peaks at the fourth service length interval and then declines slowly with the increase in service length until the seventh and last interval, but it always remained higher than the first two intervals. This result shows that the pSmad2 concentration was positively correlated with increasing of service length in the nickel smelter group. However, the concentration of pSmad2 in the officer group remains stable. Our study testifies that occupational nickel exposure could significantly increase the expression of the TGF-β signal pathway (pSmad2). 

As a regulable downstream region of the TGF-β signal pathway, *Nkx2.1* expression can be enhanced by multiple factors, such as histone post-transcriptional modification and the expression of PARP [58]. In healthy lung tissue, Nkx2.1 is first detected in ventral midgut endoderm, the expression faded along with the development of secondary bronchus and finally expressed in Terminal Respiratory Unit (TRU) [59]. After synthesis in the cytoplasm and transfer into the nucleus, it plays an important role in angiogenesis, immune response, body fluid balance and regulation of pulmonary surfactant protein by binding with specific DNA [60]. Nkx2.1 can directly or indirectly regulate the process of tumorigenesis in mature lung tissue. The reactivation and overexpression of Nkx2.1 can stimulate an early fetal gene expression pattern leading to tumorigenesis. Lung cancer patients with the overexpression of Nkx2.1 usually have a poorer prognosis in comparison to those without [14]. It can also activate the expression of p53 to decrease the transcription of LKB1, which is a critical barrier for pulmonary tumorigenesis, initiation control, differentiation and metastasis [61]. Some studies also find pulmonary surfactant can be inhibited by the interaction between Nkx2.1 and pSmad2 protein, which highly related with pulmonary immune regulation, inflammatory response and the inhibition of pulmonary malignancies deterioration (such as pulmonary surfactant protein D) [62]. Moreover, the significantly elevated expression of Nkx2.1 gene in nickel smelter group may lead to cell embryo-like proliferation and differentiation in lung tissue. Former study found that the expression and activity of OGG1 were dramatically increased in rat embryo organ, which maintains the normal development of embryo tissue [63]. However, the nickel-induced hOGG1 decreasing could further aggravate the DNA misreplication caused by frequent cell proliferation.

We observed a significantly increased expression of pSmad2 and Nkx2.1 in the nickel smelter group after 10 years of exposure. However, the concentration of pSmad2 and Nkx2.1 keep invariant in the administrative officer group. The observed results suggested that nickel exposure could continuously increase the expression of pSmad2 and Nkx2.1 independent of aging. Meanwhile, the significantly higher expression levels of pSmad2 among nickel smelter group might also enhance the expression of Nkx2.1 and especially for those who have a high dose of occupational exposure after 10 years. Considering all the samples we used in this study were from the health occupational nickel exposure population with high risk of respiratory cancer. This result may explain the high-level adjusted mortality of lung cancer in this cohort population by the increased expression of Nkx2.1 during the initiation of lung cancer development. Our canonical regression model also suggested that occupational nickel exposure could significantly increase the expression levels of Nkx2.1 and pSmad2 by the duration of occupational nickel exposure, the increased concentrations of 8-OHdG, PARP, and negatively correlated with hOGG1. 

## 5. Conclusions

Based on the evidence above, we hypothesise that the occupational nickel exposure could prominently increase the expression of Nkx2.1 and pSmad2 by oxidative damage and DNA repair changes. Meanwhile, there should be a threshold of nickel accumulation to activate the expression of the Nkx2.1 gene and pSmad2 in the human body. The transcend of this threshold will significantly increase the DNA repair and reactivate the embryonic lung development gene. The activation of embryonic lung development gene could be the trigger for the initiation of lung tumors. These results also suggest that the nickel smelters may have the most serious oxidative DNA damage and face the highest risk of lung cancer after 10–15 years of service length. To reduce the incidence and disease burden of lung cancer, strict and effective protective measures should be implemented at the beginning of 10 years of service length. Meanwhile, for the consideration that the squared model fit is 0.518 and indicates the model can only interpret the increasing expression of Nkx2.1 and pSmad2 partially. There should be other factors that can synchronously affect the activation and re-expression of embryo lung development gene, which needs further study.

## 6. Strengths and Limitations

It is the first time that we have verified a potential genetic hypothesis based on a healthy population with high risk of lung cancer that the reactivation of embryonic lung development gene and the activation of TGF-β signaling pathway could be a trigger for nickel-induced lung cancer in the initiation stage. It could provide the evidence to elucidate the potential pathogenesis and contribute to lung cancer prevention in the occupationally nickel-exposed population. Future studies with longitudinally collected samples are needed to confirm these findings. 

As limitations we should consider that all measurements were conducted in blood samples collected at baseline, and causal relationships between oxidative stress, DNA repair markers, and gene expression should be further evaluated in longitudinally collected samples. Secondly, individuals who were exposed to nickel might also be exposed to other metals and further studies are still needed to study the interaction and health effect of multi-metal exposure. 

## Figures and Tables

**Figure 1 ijerph-16-00120-f001:**
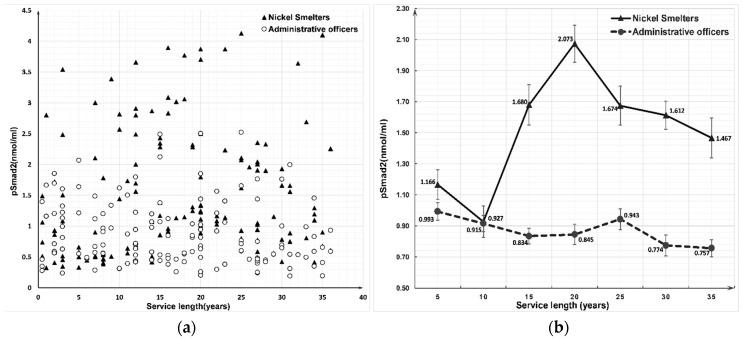
Scatterplot (**a**) and tendency (**b**) chart of pSmad2 concentration between nickel smelters and administrative officers at different service length.

**Figure 2 ijerph-16-00120-f002:**
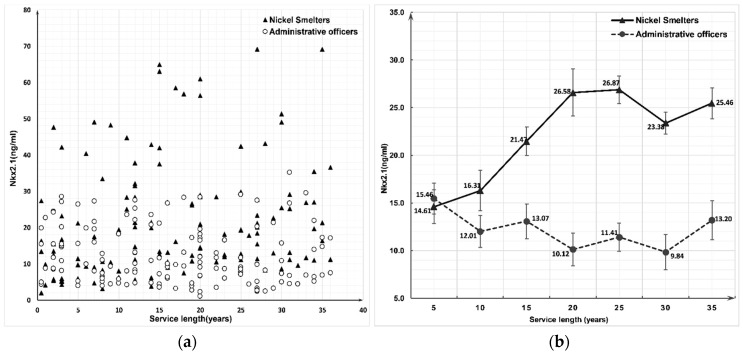
Scatterplot (**a**) and tendency (**b**) chart of Nkx2.1 concentration between nickel smelters and administrative officers at different service length.

**Table 1 ijerph-16-00120-t001:** Urinary nickel concentrations among different occupational classification groups in Jinchang cohort.

Gender	Nickel Exposure Level (μg/L Creatinine)	*n*	Mean (95% CI)	St.d *	Selected Percentiles					Kruskal-Wallis	
					5th	25th	50th	75th	95th	H	*p*
Male	Low (Administrative stuff)	25	4.61 (3.27–5.94)	0.65	1.62	2.33	3.43	5.87	12.96	8.767	0.003
	Medium (Technician)	125	6.88 (5.28–8.48)	0.85	1.79	3.15	5.10	7.28	16.39		
	High (Smelter & Miner)	101	8.75 (7.44–10.05)	0.66	2.08	4.43	6.56	12.23	21.32		
	Sum	251	7.40 (6.44–8.37)	0.51	1.79	3.24	5.27	9.12	19.13		
Female	Low (Administrative stuff)	25	6.12 (3.60–8.65)	1.22	2.10	3.14	4.31	6.04	24.16	0.688	0.407
	Medium (Technician)	125	7.88 (6.25–9.52)	2.24	2.09	3.72	6.12	8.88	20.02		
	High (Smelter & Miner)	99	12.65 (8.99–16.32)	2.04	1.93	3.63	5.91	12.97	43.66		
	Sum	249	9.61 (7.90–11.31)	1.40	2.05	3.59	5.78	9.28	27.52		

Note: * St.d represents the standard deviation.

**Table 2 ijerph-16-00120-t002:** Baseline characteristics of the 280 participants.

Characteristic	Categories	Nickel Smelters		Administrative Officers	
		*n* = 140	%	*n* = 140	%
Gender	Male	140	100	140	100
	Female	0	0	0	0
Age	20–24	2	1.4	2	1.4
	25–29	29	20.7	29	20.7
	30–34	26	18.6	23	16.4
	35–39	23	16.4	26	18.6
	40–44	20	14.3	20	14.3
	45–49	20	14.3	20	14.3
	50–55	20	14.3	20	14.3
Nationality	Han nationality	138	98.6	140	100
	Man nationality	1	0.7	0	0
	Other	1	0.7	0	0
Occupation	Cadre	0	0	63	45
	Worker	140	100	3	2.1
	Technician	0	0	7	5
	Service	0	0	67	47.9
Marital status	Not married	17	12.1	21	15
	Married	118	84.3	118	84.3
	Other	5	3.6	1	0.7
Residence years	<20	17	12.1	32	22.9
	20–39	100	71.4	79	56.4
	>40	23	16.4	29	20.7
Education status	Primary and below	4	2.9	1	0.7
	Junior high school	34	24.3	6	4.3
	Senior high school	52	37.1	22	15.7
	Undergraduate and above	50	35.8	111	79.3
Family income (RMB/month/person)	<1000	10	7.1	10	7.1
	1000–2000	80	57.1	34	24.3
	≥2000	50	35.7	96	68.6
Cancer family history	Non	106	75.7	101	72.1
	Case	32	22.9	38	27.1
	Unknown	2	1.4	1	0.7

**Table 3 ijerph-16-00120-t003:** Distribution of behavioral factors in nickel smelters and administrative officers.

Behavioral Factors	Categories	Nickel Smelters		Administrative Officers		Chi-sq *	*p*
		*n* = 140	%	*n* = 140	%		
Smoking	Non	43	30.71	46	32.86	0.37	0.83
	Past	18	12.86	15	10.71		
	Current	79	56.43	79	56.43		
Smoking index	0–400	108	77.10	120	85.70	3.4	0.07
	>400	32	22.90	20	14.30		
Alcohol drinking	Non	84	60.00	76	53.60	1.80	0.41
	Past	9	6.40	7	5.00		
	Current	47	33.60	58	41.40		
Drinking index	0	85	60.71	77	55.00	1.49	0.68
	<5200	32	22.86	34	24.29		
	5200–10,400	10	7.14	15	10.71		
	>10400	13	9.29	14	10.00		
Tea consumption	Non	42	30.00	44	31.40	0.2	0.91
	Past	4	2.90	3	2.10		
	Current	94	67.10	93	66.40		
Tea index	0	42	30.00	45	32.10	1.19	0.76
	<600	55	39.30	60	42.90		
	600–1200	21	15.00	18	12.90		
	>1200	22	15.70	17	12.10		

Note: * Chi-sq represents Chi-Square test.

**Table 4 ijerph-16-00120-t004:** Expression of oxidative damage (MDA) and DNA repair (PARP) in different service length categories between nickel smelters group and administrative officers group.

Service Length (Years)	MDA (nmol/mL)				PARP (ng/mL)			
	Nickel Smelters		Administrative Officers		Nickel Smelters		Administrative Officers	
	*n*	*x* ± st.d *	*n*	*x* ± st.d	*n*	*x* ± st.d *	*n*	*x* ± st.d
≤4	20	11.3668 ± 5.0144	20	6.1012 ± 3.8763	20	0.0405 ± 0.0017	20	0.0402 ± 0.0010
5–9	20	13.7679 ± 6.1174	20	7.5872 ± 4.2609	20	0.0406 ± 0.0015	20	0.0398 ± 0.0008
10–14	20	14.8466 ± 3.6641	20	9.6055 ± 5.9680	20	0.0414 ± 0.0028	20	0.0402 ± 0.0013
15–19	20	16.0179 ± 4.6807	20	10.8349 ± 4.8698	20	0.0405 ± 0.0012	20	0.0401 ± 0.0012
20–24	20	17.6983 ± 2.9618	20	13.7706 ± 5.7772	20	0.0405 ± 0.0021	20	0.0394 ± 0.0002
25–29	20	17.1786 ± 3.0584	20	14.5487 ± 4.7357	20	0.0399 ± 0.0004	20	0.0397 ± 0.0009
≥30	20	21.5776 ± 29.7619	20	15.0885 ± 8.9044	20	0.0397 ± 0.0006	20	0.0396 ± 0.0006
Total	140	16.0648 ± 12.0839	140	11.0766 ± 6.4684	140	0.0405 ± 0.0017	140	0.0398 ± 0.0009
*Z* (Wilcoxon)	6.125				4.853			
*p*	<0.001				<0.001			
*F* (ANOVA)	1.47		7.644		2.217		2.255	
*p*	0.193		0		0.045		0.042	
*r*_s_ (Spearman)	0.273		0.569		−0.175		−0.376	
*p*	<0.001		<0.001		0.039		<0.001	

Note: * *x* represents the mean of MDA concentration. St.d represents the standard deviation.

**Table 5 ijerph-16-00120-t005:** Expression of NKx2.1 and pSmad2 in the different service length categories between nickel smelters group and administrative officers group.

Service Length (Years)	Nkx2.1 (ng/mL)				pSmad2 (nmol/mL)			
	Nickel Smelters		Administrative Officers		Nickel Smelters		Administrative Officers	
	*n*	*x* ± st.d *	*n*	*x* ± st.d	*n*	*x* ± st.d	*n*	*x* ± st.d
≤4	20	14.6062 ± 12.3940	20	15.4643 ± 7.3406	20	1.1658 ± 0.8780	20	0.9929 ± 0.5046
5–9	20	16.3074 ± 14.4587	20	12.0121 ± 7.5392	20	0.9272 ± 0.9011	20	0.9149 ± 0.4639
10–14	20	21.4685 ± 12.4842	20	13.0714 ± 8.1059	20	1.6796 ± 1.0290	20	0.8335 ± 0.4476
15–19	20	26.5803 ± 19.9658	20	10.1228 ± 7.6858	20	2.0731 ± 0.9835	20	0.8448 ± 0.5806
20–24	20	26.8712 ± 15.4844	20	11.4143 ± 6.6340	20	1.6744 ± 1.0137	20	0.9429 ± 0.5985
25–29	20	23.3837 ± 14.1935	20	9.8430 ± 8.2613	20	1.6122 ± 0.8516	20	0.7739 ± 0.5957
≥30	20	25.4599 ± 16.2387	20	13.198 ± 9.1051	20	1.4666 ± 1.0241	20	0.7567 ± 0.4891
Total	140	22.0967 ± 15.5792	140	12.1608 ± 7.8845	140	1.5141 ± 0.9988	140	0.8657 ± 0.5240
*Z* (Wilcoxon)	−6.123				−5.845			
*p*	<0.001				<0.001			
*F* (ANOVA)	2.099		1.242		3.069		0.552	
*p*	0.057		0.289		0.008		0.768	
*r*_s_ (Spearman)	0.312		−0.146		0.232		−0.169	
*p*	<0.001		0.085		0.006		0.046	

Note: * x represents the mean of Nkx2.1 and pSmad2 protein concentration. St.d represents the standard deviation.

**Table 6 ijerph-16-00120-t006:** Canonical correlations between embryonic lung development gene and potential influence factors in nickel smelters.

**Group V**	**Canonical Coefficient**	
	***χ*^1^**	***χ*^2^**
Service length	0.2784	0.875
MDA	0.0369	0.1734
8-OHdG	0.764	−0.6255
PARP	0.5858	−0.2695
hOGG1	−0.4531	−0.0561
Proportion explained by own var %	28.39	25.26
**Group W**	***η*^1^**	***η*^2^**
Nkx2.1	0.9886	0.1507
pSmad2	−0.0083	1
Proportion explained by own var %	48.87	51.13
*R* ^2^	0.518	0.033
*R*	0.72	0.18
Adjusted *R*	0.709	0.124
*p*	<0.0001	0.347

Note: *χ* and *η* represents the canonical correlation coefficient 1 and 2 for group V and W. *R*, *R*^2^, adjust *R* represents the probability that Group W correlated with Group V in canonical coefficient 1 and 2. *p* represents the probability value of different pairs of canonical variables.

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
