# Peer review of "The Relationship between Nkx2.1 and DNA Oxidative Damage Repair in Nickel Smelting Workers: Jinchang Cohort Study"

_ijerph, 2019, doi:10.3390/ijerph16010120_

Reviewer 1 Report

In this study the authors assess putative differences of selected biomarkers between nickel smelters and administrative officers in the Jinchang Cohort and attempt to correlate Nkx2.1 expression levels with DNA oxidative damage repair.

1. The manuscript contains many grammar and syntax errors that need to be corrected (for examples, please refer to the attached file).

2. In addition, in some cases methods are not adequately described or some results are not clear to the reader (please refer to the attached file).

3. Most importantly, part of the study has been already published in reference 25 (i.e. the differential levels of 8-oHdG and hOGG1 between smelters and administrative officers have been reported). Since this data is redundant to a previous study, experiments concerning these markers should be eliminated from the current study and information from the previous study should be discussed and related to the new data presented here in the Discussion section.

Author Response

Dear reviewer:

Sincerely appreciate for the elaborate and professional comments and suggestions from you which gave us a great help during the manuscript revision. Detailed modifications were made based on your comments, which were listed as follow:

1. All grammar and syntax errors in the previous version have been completely revised by an English native speaker. The revised manuscript was attached with this mail.

2. Some methods and results parts were also revised as your comments (please refer to the attached file)

    Among all of your comments, there are two of them I'd like to make a further explanation.

2.1. About the previously published results of 8-OHdG and hOGG1:

    We belong to the same research team with those authors who previously published the results about 8-OHdG and hOGG1. We have done the experiment together and shared  all the research results about oxidative damage, DNA repair, and the embryonic lung development gene, etc. Any of the 8-OHdg and hOGG1 was indispensable to make an integrated and logical understanding for the reviewers and readers about our works. So, I eliminated these results about 8-OHdG and hOGG1 from the results part, discussed and related to the Discussion section. The detail results were presented as supplemental materials.   

2.2. About the positive & negative error bars in figure 2: I re-checked the original data found that the negative error values were accidentally linked to inaccurate data. Both figure 1 and figure 2 were re-created and updated in the revised manuscript.

Best regards and really appreciated for your help and suggestions!

Reviewer 2 Report

This study investigated the influence of occupational nickel exposure on oxidative stress (MDA and 8-oHdG), DNA repair (hOGG1 and PARP), pSmad2 and Nkx2.1 expression in the blood samples of 140 nickel smelters and 140 age-matched administrative officers. The results showed that the concentrations of MDA, 8-oHdG, PARP, pSmad2, and Nkx2.1 were significantly increased, while the hOGG1 was decreased in nickel smelters. Nkx2.1 and pSmad2 were positively correlated with the employment length in nickel smelters but not in the administrative officer group. Furthermore, the increased expression of Nkx2.1 and pSmad2 was correlated with the nickel-induced oxidative damage and DNA repair change. There are some concerns about the inconsistency and typos as listed in the following.

L21: x2 test->χ2 test

L22, 23: 8-oHdG vs. L63: 8-OHdG vs. L102: 8-Ohdg (check all)

L127: minutes vs. min

*L162-: Table 4, Table 5, Table 6: The data expression must consider the significant figure.

L162-: P(Wilcoxon), P(ANOVA), P(Spearman)

L155: group. (Table 4)

L165, L234-246: hoGG1 vs. hOGG1 (check all)

L175: Nkx2.1 and pSmad2 (check all, gene vs. protein)

L176, 177: P<0.001< span="">

*L228: No correlation between service length and DNA oxidative damage was observed in both nickel smelter group and officer group. ->But, L152-154: [the concentration showed a significant decreasing trend with increasing employment length (rs=-0.239, p=0.005) among the nickel smelter group,]

L250-:OGG1 vs. oGG1

L305: Secondary bronchus

*L332-334: occupational nickel exposure could significantly increase the expression levels of Nkx2.1 and pSmad2 by the duration of occupational nickel exposure, the concentrations of 8-oHdG, PARP, and hOGG1. -> But, hOGG1 is decreased in nickel smelters [Table 5]

L339: PSmad2

*L380: References: keep the consistent format for title (R10, 11, etc.: capital letter) and journal (full name vs. abbreviation)

Author Response

Dear reviewer:

Sincerely appreciate for the elaborate and professional comments and suggestions from you which gave us a great help during the manuscript revision. 

Detailed modifications were made based on your comments, which were listed as follow:

1. All grammar and syntax errors in the previous version have been completely revised by an English native speaker. The revised manuscript was attached with this mail.

2. Some methods and results parts were also revised as your comments (please refer to the attached file).

3. All the concerns about the inconsistency and typos you listed were carefully checked and corrected (please refer to the attached file).

About all the comments there are two of them I'd like to make a further explanation:

1. "L228 you mentioned in the previous version": I made a mistake when present the results about the correlation between DNA oxidative damage and service length. You are correct that only the administrative officers group showed no correlation between service length and DNA oxidative damage. Based on the comments from another reviewer, I moved this part to the discussion section, which showed at the Line 258-259 in the revised manuscript (please refer to the attached file).

2. "L332-334 you mentioned in the previous version": we only observed that occupational nickel exposure could decrease the expression level of hOGG1 when compared with administrative officers group. Also, no correlation was observed between the expression level of hOGG1 and service length in both nickel smelter group and administrative officer group. Somehow, the canonical regression model showed a comprehensively and statistically positive correlation between the expression level of hOGG1 and Nkx2.1 in the previous version. 

   So we re-checked the model and found that in the previous version we imported some confounding factors in the canonical model such as age which resulted in an incorrect result. After eliminating those error, the model showed a logical result which consisted with preceding results. However, for the consideration that the squared model fit is 0.518 and indicates the model can only interpret the increasing expression of Nkx2.1 and pSmad2 partially. There should be other factors that can synchronously affect the activation and re-expression of embryo lung development gene, which needs further study.

3. All the references were re-formatted based on the template that MDPI (Molecular Diversity Preservation International) provided.

Best regards and really appreciated for your help and suggestions!

Round  2

Reviewer 1 Report

This is the revised version of a previously submitted manuscript.

The authors have addressed all of my comments during the previous reviewing round.

Author Response

Dear reviewer

     Sincerely grateful for your suggestions and help!

     Best regards and many thanks!